# Value Positions and Relationships in the Swedish Digital Government

**Leif Sundberg**

Department of Information Systems and Technology, Mid Sweden University, Sundsvall 851 70, Sweden;
leif.sundberg@miun.se

**Abstract:** Governments across the world spend vast resources on implementing digital technology. Electronic, or digital, government is the use and study of Internet-based information and communication technology in the public sector. A point of departure in this study is that investments in technology are not value-free; they require allocation of limited resources and trade-offs between values. The purpose of this paper was to investigate how values are prioritized in the Swedish digital government. This research was conducted by using quantitative data from a survey administered to Swedish municipalities and national agencies. In addition, qualitative data from a database was used to exemplify value operationalization. The research utilized a theoretical framework based on four value positions: professionalism, efficiency, service, and engagement. The findings reveal that service and quality, and productivity and legality have a high priority, while engagement values are less prioritized. Differences based on organization type and size are also discussed. Moreover, the study suggests that professionalism and efficiency are distinct value positions, while service and engagement are closely related through citizen centricity. The qualitative material suggests that citizen centricity can manifest itself as a form of service logic, but also in the form of educational digital inclusion activities for vulnerable groups. The paper concludes by suggesting that future research should further refine the concept of citizen centricity in relation to digital government values, since its current meaning is ambiguous.

**Keywords:** digitalization; e-Government; public values

## 1. Introduction

Electronic, or digital, government, hereinafter e-Government, is the use of Internet-based information and communication technology (ICT) in the public sector. Bannister and Connolly (2014) point out that the implementation of ICT is not value-free; it requires decisions about—and sometimes trade-offs between—values. The public manager needs to prioritize the allocation of limited resources that have value in their alternative uses. Based on the high failure rate of e-Government initiatives, Flak et al. (2009) propose that researchers should utilize a structured approach to benefit realization, combined with a focus on values. Rose et al. (2015b) argue that public sector information technology (IT) initiatives with multiple stakeholder groups may benefit from working with values during design and evaluation. Furthermore, these authors claim that studying values might help expose empty rhetoric in the formulation of e-Government objectives.

Values in e-Government have been previously studied in various contexts. Although there are variances in the definition of what exactly constitutes these values, some attempts to define them can be found in the literature. According to Bozeman (2009), public values can be described as the normative consensus about rights, obligations and principles between the citizen and the government. Bannister and Connolly (2014) define public values as modes of behavior that are believed to be right.

Rose et al. (2015a) reveal that Danish public managers prioritize administrative efficiency while neglecting citizen empowerment values. Another study from Denmark showed that, while professionalism, efficiency, and service values were relatively stable in government IT strategies produced between 1994 and 2016, engagement values declined (Stouby-Persson et al. 2017). Ilshammar et al. (2005) describe how Swedish policy documents often mention the promotion of democratic processes in relation to technology, but when these processes are operationalized, rationalization and efficiency are prioritized.

Against this backdrop, the purpose of this paper is to investigate how values are prioritized in the Swedish digital government. Three hypotheses are formulated:

**H1:** *Value priorities differ based on organization type (national agency / municipality).*

(H0: There are no differences based on organization type.)

**H2:** *Value priorities differ based on organization size*

(H0: There are no differences based on organization size.)

**H3:** *Values can be divided into four distinct positions (professionalism, efficiency, service, and engagement, see Section 2).*

(H0: Values cannot be divided into four distinct positions.)

This research was based on a nationwide survey administered to Swedish municipalities and national agencies. Qualitative data from a database was used to gather examples of value operationalization. The novelty of this approach was the combination of these two datasets to generate additional understanding of public values from two levels of government (local and national), as well as to test prior theories about public sector values.

This paper proceeds as follows: In Section 2, value positions in the public sector are presented, followed by a brief description of values in the Swedish digital government context (Section 3). The theory from Section 2 was used to construct a survey, which is described in Section 4, Materials and Methods. In Section 5, the results are presented. Section 6 contains conclusions, limitations and directions for future research.

## 2. Theoretical Background: Value Positions

Different paradigms in the public sector have replaced and advanced the roles of citizens, policymakers and government administrators. The expected value outputs also differ between paradigms. Value positions can be congruent or divergent, e.g., an increased focus on one position might lead to more, or less, focus on another position.; they can support each other, or be in a state of conflict. Persson and Goldkuhl (2010) argue that e-Government values are a merge between values from Weberian bureaucracy and New Public Management (NPM). Andersen et al. (2012) derived seven value dimensions (the public at large, rule abidance, balancing interests, budget keeping, efficient supply, professionalism, and user focus) from a survey of public managers in Denmark. These authors also found differences depending on organizational levels and tasks. Van der Wal et al. (2008) discussed a 'common core' of values that are important in both the public and the private sector (accountability, expertise, reliability, efficiency and effectiveness). Bannister and Connolly (2014) distinguish between duty-oriented, service-oriented, and socially-oriented values associated with ICT in the public sector.

For this research, a framework of value positions by Rose et al. (2015b) was utilized to create a survey. These authors presented a value classification based on paradigms of managerial work in the public sector that differentiates between:

- Professionalism values (e.g., acting according to laws and regulations)

- Efficiency values (e.g., cost savings, performance)
- Service improvement values (e.g., used-needs based approaches)
- Citizen engagement values (e.g., including citizens in policy and decision making)

This framework has two advantages: First, it has a solid theoretical foundation (as requested by Rutgers, 2008) in managerial value paradigms, as will be further developed below. Second, it discusses the role of e-Government and IT in relation to these paradigms.

Table 1 summarizes the value positions, their origins, and representative values identified by Rose et al. (2015b). This framework has been utilized in prior research. For example, by combining this framework with stakeholder theory, Rose et al. (2018) studied three cases in the Norwegian context. Stouby-Persson et al. (2017) applied it to policy documents from the Danish e-Government, as mentioned in the introduction.

**Table 1.** Value positions.

| Value Position | Paradigm | Representative Values |
|---|---|---|
| Professionalism | Weberian bureaucracy | Durability, equity, legality, and accountability. |
| Efficiency | New public management | Value for money, cost reduction, productivity, and performance. |
| Service improvement | Public value theory | Commitment to the public interest expressed through public service, citizen centricity, service level and quality. |
| Citizen engagement | New public service | Democratic engagement, deliberative engagement and participative engagement. |

These categories will be used to construct a survey, which is described in Section 4. This section continues with a more detailed outline of the respective paradigms.

*2.1. Professionalism*

In traditional Weberian governments, rules, due process, and neutrality are the core values that should determine how the public sector acts. The public manager is a rational-legal authority limited by its sphere of competence within a hierarchical organization that builds on fixed areas of activities and division of labor. The bureaucratic organization is superior to other forms of organization: Weber compared the bureaucratic apparatus superiority with a machine's superior production ability over non-mechanical modes of production (Weber [1922] 1968). The bureaucracy is independent, robust, and consistent, governed by rule of law, where the public record is the basis of accountability. The role of e-Government, according to the professionalism ideal, is to provide a flexible and secure digital public record to support standardized administrative procedures; IT constitutes an information infrastructure that enacts the regulatory system (Rose et al. 2015b).

*2.2. Efficiency*

Weberian bureaucracy dominated much of the twentieth century but was questioned after the economic (oil) crisis of the 1970s. In the 1980s, a new paradigm that is closely connected to the market economy appeared: new public management (NPM). According to NPM, Weberian bureaucracy failed to answer customer needs, which led to under performance and poor legitimacy. The dominating core value of NPM is efficiency: The public administration is slim and efficient, minimizing the waste of public resources. The citizen is seen as a customer whose demands can be satisfied by proper government supply. Prior ideals in the public sector suggested that accountability could be increased, and corruption could be reduced by separating the private and public sectors. In NPM, the distinction between these two sectors is removed and accountability is achieved through obtaining results measured in monetary terms. Furthermore, the ideal organizational structures are

small competing units, inspired by private sector corporations (Hood 1991). IT is associated with automation, and is considered a tool for productivity that substitutes labor (Rose et al. 2015b). Although NPM is often associated with Margaret Thatcher's United Kingdom and Ronald Reagan's United States, Hood (1995) pointed out that Sweden heavily emphasized NPM in the 1980s.

### 2.3. Service

The main criticism of NPM is its emphasis on efficiency by copying features from the private sector. Moore (1995) argues that in the private sector the individual can refrain from consuming a product whose value is perceived as limited, while in the public sector the government uses its coercive power of taxation to produce services that may be mandatory for individuals. The challenge for the public manager is to identify which consequences will produce public value. Alford and Flynn (2009) argue that public value can be deployed as both an empirical theory of what public managers do and normative prescriptions of what these managers should do. Cook and Harrison (2015) concluded that public value analysis may be beneficial for identifying internal and stakeholder values to improve an agency's change management and communication strategies. An E-government's role in relation to this ideal is to produce online services. IT is seen as a service enabler, increasing access and quality of services (Rose et al. 2015b). Dunleavy et al. (2006) use the term "digital-era governance" to describe this paradigm shift and identify three characteristic themes: reintegration (as opposed to fragmentation), needs-based holism (i.e., reorganization to create seamless, non-stop solutions) and digitization processes (electronic service delivery).

### 2.4. Engagement

The engagement ideal builds on the idea of actively engaging citizens through participatory processes. Based on liberal democratic ideas, civil society stakeholders are expected to participate in, for example, policy development. Social networks are one example of the types of technologies that may facilitate such engagement values (Rose et al. 2015b). In this paradigm, sometimes termed new public service, governance is based on democratic citizenship and community. The primary role of the public servant in the engagement paradigm is to help citizens articulate and meet their shared interests rather than attempt to control or steer society (Denhardt and Denhardt 2000). In e-Government research, engagement processes facilitated by technology are studied within the e-Participation subfield (Sæbø et al. 2008).

## 3. Values in the Swedish e-Government Context: A Short History

Computerization of the Swedish public sector is a popular study subject, especially in political science (see e.g., Ilshammar et al. 2005; Lundin 2008, 2015). After World War II, the Swedish public sector was expanding and the hopes were that computers would contribute to reduced costs. Several large registers resided in the Swedish public sector, including a population register of all of the nation's individuals. Over the next decades, these registers were computerized using automatic (or administrative) data processing (ADP). The expectations of what this new technology could achieve were high, and government investigations identified several areas of application (Finansdepartementet 1962). Hence, during this period, computing in the Swedish public sector can was motivated by efficiency through cost savings, congruent with professionalism, manifested by ADP as a tool to carry public registers.

During the 1970s, both economic growth and optimism surrounding computers halted in Sweden. Professionalism in the form of legality started to act as a convergent constraint to efficiency, by reducing the inter-organizational information flow in the public sector. Concerns about computer-related issues such as integrity, security and work environment threats led to increased demands for political control over this technology. One manifestation of this control was the world's first computer law. This law essentially required government agencies to apply for a permit to create a register and allowed individuals to request a copy of the information about themselves from the registers (Justitiedepartementet 1973). The technology changed, during this time, from large central computers

to decentralized desktop computers suitable for office use. Political control over computers was reduced during the 1980s, as computer investments were meant to be up to each agency, in line with new management ideals.

In the 1990s, the optimism associated with new technology, IT, returned. A speech by the Swedish Prime Minister in 1994 laid out a path with a clear goal: Sweden should be a leading IT nation no later than 2010. For the public administration, this goal meant adapting Internet-based technology into a "24-h agency." The documents suggest a congruence of values: By adopting Internet-based technology, public entities would become more efficient, increase services via web sites, and facilitate engagement processes (see e.g., IT-kommissionen 1994; Regeringen 2000).

After the millennium shift, which was characterized by a declining IT industry during the dotcom crisis, the effects of computerized technology in the public sector were questioned. A final report from the "democracy investigation" summarized the then-current initiatives as mostly part of a "service democracy," whereby information from politicians was supplied to civil society and not used as a tool for active engagement. Concerns were also raised about unequal access to and usage of technology. Younger, well-educated people with high incomes tended to use the Internet more than other groups; also, more men than women utilized the Internet (Demokratiutredningen 2000). In 2004, the National Audit Office concluded that efforts made since the 1990s to establish a 24-h government had had limited effects. Due to a significant focus on cost saving, few advanced services targeting smaller user groups had been created. The use of e-mail among government entities was described as a threat to the rule of law, and the work to remove legal obstacles obstructing the implementation of e-services was reported as slow (Swedish National Audit Office 2004). Once again, optimism regarding technology had been replaced by pessimism, and value congruence was replaced by value conflicts in evaluation reports.

In subsequent years, the service ideal received additional attention, especially in the form of Internet use for e-services. Action plans and strategies emphasized the ease of use (Regeringskansliet 2008) and citizen centricity (Regeringskansliet 2012). The title IT Minister was converted to Digitalization Minister in 2016. By intensifying the use of a new range of technologies, such as big data, artificial intelligence and Internet of Things, once again the hopes of value congruence enabling by technology (now referred to as "digitalization") were raised. By adopting laws and regulations, services were meant to be "digital by default" and the administration more efficient by, for example, using automated decision making (see e.g., Digitaliseringskommissionen 2016; Utredningen om effektiv styrning av nationella digitala tjänster 2017).

## 4. Materials and Methods

This section contains descriptions of the data (Section 4.1) and the procedure (Section 4.2).

### 4.1. Data Description

This research was performed by administering a survey to all Swedish municipalities (*n* = 290) and national government agencies (*n* = 228). The survey was sent to each government body's official e-mail address and asked for a respondent who was responsible for overall digitalization (such as a coordinator, strategist, or decision maker). Background variables were:

- Organization type (municipality or national agency)
- Organization size: number of inhabitants (municipality) or number of employees (agencies)

The survey consisted of eight value propositions based on the aforementioned classification by (Rose et al. 2015b) (Section 2). Two values from each position were chosen based on how easily they could be translated into Swedish. Sometimes the value itself was used in the survey (e.g., productivity), while others had to be expressed through a sentence (e.g., accountability). Table 2 includes the values and their definitions, and the survey can be found in Appendix A.

**Table 2.** Values and definitions.

| Value Position | Value Definitions (Rose et al. 2015b) |
|---|---|
| Professionalism | **Legality**: Framing decisions by laws and authorized policies.<br>**Accountability**: Traceable responsibility for legitimate actions in a chain of command, documented in the public record. |
| Efficiency | **Cost reduction:** Reduce cost per output unit.<br>**Productivity:** Increase output unit per economic unit. |
| Service | **Service level and quality:** Provision of services, which meet citizens' expectations.<br>**Citizen centricity:** Respect for each citizen's individual interests. |
| Citizen engagement | **Participative engagement:** Engaging civil society in decision making.<br>**Democratic engagement:** Engaging civil society in democratic processes. |

In addition, two values related to the use of technology were added. Each of the 10 values was then transformed into a sentence; for example:

We digitalize to …
… keep up with technology
… be at the forefront of using technology.

The respondents were asked to grade each value on a Likert scale from 0 to 10, where 0 meant not prioritized at all and 10 meant highest priority. A relatively high number of items on the scale were used to reduce the number of "uncertain" answers, which occurred when respondents used the middle category by default (Matell and Jacoby 1972). The survey was created in Google Docs. Confidentiality of the answers was promised. The values were randomly presented to the survey respondent to ensure their order would not affect the answers. The survey was open to answers for 21 days. The total number of respondents was $n = 240$, of which 127 were from municipalities (43.8% of all municipalities) and 113 were from national government agencies (49.6% of all national agencies).

While it should be acknowledged that two values from each position is rather sparse, it is important to maintain a satisfying subject to item ratio in the subsequent analysis. While a ratio of 10:1 is a common rule of thumb, even at ratios of 20:1 (20 samples per item), principal component analysis can produce error rates up to 30%. Osborne and Costello (2004) argue that the researcher should apply a "more is always better" approach to sample size rather than aiming for a critical ratio. The sample size in this study gives a ratio of 1:24 with respect to the ten items mentioned above, and 1:30 if only the eight items from the four value positions are included.

Using the categories in Table 3, the sizes of the national agencies were classified depending on the number of employees (the Swedish Agency for Public Management 2018).

**Table 3.** Size of national agencies.

| Number of Employees | *n* |
|---|---|
| Small ≤50 | 17 |
| Medium 51–999 | 79 |
| Large ≥1000 | 17 |

Using the intervals in Table 4, the municipality sizes were classified depending on their population (SKL 2014).

**Table 4.** Size of municipalities.

| Municipality Population | *n* |
|:---:|:---:|
| <10,000 | 23 |
| 10,000–19,999 | 51 |
| 20,000–49,999 | 31 |
| 50,000–99,999 | 15 |
| ≥100,000 | 7 |

An analysis of the responses showed that mainly the smallest national agencies refrained from answering the survey. The same was true of the municipalities, where the smaller entities are slightly less represented compared to the actual population.

*4.2. Procedure*

The analysis was performed with IBM's SPSS 23 software. Prior research has shown that ordinal and Likert scales are often empirically linear and can thus be treated as intervals (see e.g., Carifio and Perla 2007, 2008; Norman 2010); in this study, the responses from the survey have, therefore, been treated as an interval scale. The statistical tests are summarized in Table 5, and further elaborated below.

An independent *t*-test (95% confidence interval) was utilized to compare how municipalities and national agencies prioritized the values (H1). One-way ANOVA tests were used to compare the means of value prioritization based on organization size (H2). Pearson's correlation was used to estimate correlations between the values. Then, a principal component analysis (Eigenvalue > 1, Varimax with Kaiser Normalization) was performed on the eight values based on Rose et al. (2015b). According to the H3, the values should represent four dimensions (as described in Table 2).

A final step in the research was conducted to find examples of operationalization of the different values. This step required an interpretative approach, where initiatives from the "Dela digitalt" (Eng: Digital sharing) (Dela Digital 2018) website were analyzed. In the database, Swedish government entities can exchange knowledge and present their work with digitalization. The analysis was performed by searching for terms such as "service", "democracy", "efficiency" and "law" in the database, as well as browsing the content. Then, the results aided the interpretation of how government entities work with professionalism, legality, service, engagement and technology. This step in the research was not subject to quantitative measurement, but served as examples of how the values are realized in practice.

**Table 5.** Statistical tests.

| Test | Purpose | Results |
|:---:|:---:|:---:|
| Descriptive statistics | Display value prioritizations. | Tables 6–8 |
| Independent *t*-test | Compare means between municipalities' and national agencies' value prioritization (H1) | See paragraph above Table 6 |
| One-way ANOVA | Compare means based on organization size (H2) | Table 9 |
| Pearson's test | Show correlation between values. | Table 10 |
| Principal component analysis | Reduce the number of dimensions (H3) | Tables 11–14 |

## 5. Results

Section 5 is divided into three parts. In Section 5.1, descriptive statistics are presented, together with differences based on organization type and size. In Section 5.2, correlations and the results of the factor analysis are presented. Finally, examples of operationalizations are described in Section 5.3.

*5.1 Prioritization of Values*

Table 6 shows the overall results, while Tables 7 and 8 showcase the data of the national agencies and the municipalities, respectively. Service and quality were highly ranked by all government entities, together with productivity and legality. Independent *t*-tests showed that the municipalities graded all values except for professionalism and technocratic values significantly higher ($p > 0.01$) than the national agencies. The results of the test are reflected in the descriptive statistics which suggest that municipalities adopted a more citizen-centric approach. National agencies prioritized the engagement values lowest. The national agencies had higher standard deviations across all values. One-way ANOVA tests showed no significant differences in means between the municipalities, but were dependent on population size. Among the national agencies, significant differences were found concerning efficiency and technocratic values. The largest agencies had the highest means in the efficiency category, while medium and large agencies prioritized technology higher than the small agencies.

**Table 6.** Descriptive statistics, overall results.

| Value | Mean | Median | Std. Dev |
|---|---|---|---|
| Service and quality | 8.58 | 9 | 1.928 |
| Productivity | 7.97 | 8 | 1.747 |
| Legality | 7.75 | 8 | 2.299 |
| Citizen centricity | 7.30 | 8 | 2.677 |
| Cost reduction | 6.87 | 7 | 2.010 |
| Keep up with technology | 6.76 | 7 | 2.014 |
| Democratic engagement | 6.27 | 7 | 2.829 |
| Accountability | 5.65 | 6 | 2.480 |
| Participative engagement | 5.56 | 6 | 2.812 |
| Technological forefront | 5.38 | 5 | 2.397 |

**Table 7.** Descriptive statistics, national agencies.

| Value | Mean | Median | Std. Dev |
|---|---|---|---|
| Service and quality | 7.89 | 8 | 2.396 |
| Productivity | 7.62 | 8 | 2.080 |
| Legality | 7.72 | 8 | 2.351 |
| Keep up with technology | 6.64 | 7 | 2.260 |
| Cost reduction | 6.35 | 7 | 2.145 |
| Citizen centricity | 6.18 | 7 | 3.183 |
| Accountability | 5.35 | 5 | 2.685 |
| Technological forefront | 5.30 | 5 | 2.705 |
| Democratic engagement | 5.09 | 5 | 3.184 |
| Participative engagement | 4.24 | 5 | 2.974 |

**Table 8.** Descriptive statistics, municipalities.

| Value | Mean | Median | Std. Dev |
|---|---|---|---|
| Service and quality | 9.20 | 10 | 1.069 |
| Citizen centricity | 8.31 | 8 | 1.561 |
| Productivity | 8.28 | 8 | 1.319 |
| Legality | 7.78 | 8 | 2.260 |
| Cost reduction | 7.33 | 7 | 1.764 |
| Democratic engagement | 7.31 | 8 | 1.955 |
| Keep up with technology | 6.87 | 7 | 1.768 |
| Participative engagement | 6.73 | 7 | 2.045 |
| Accountability | 5.93 | 5 | 2.258 |
| Technological forefront | 5.46 | 5 | 2.092 |

**Table 9.** Differences based on organization size (national agencies).

| Org.size | Value | Mean | Median | Std. Dev |
|---|---|---|---|---|

| | | | | |
|---|---|---|---|---|
| Small | Productivity | 6 | 7 | 3.298 |
| Medium | Productivity | 7.71 | 8 | 1.642 |
| Large | Productivity | 8.82 | 9 | 1.380 |
| Small | Cost reduction | 5.88 | 7 | 2.571 |
| Medium | Cost reduction | 6.16 | 6 | 1.99 |
| Large | Cost reduction | 7.65 | 8 | 2.029 |
| Small | Keep up w. tech | 4.94 | 5 | 2.277 |
| Medium | Keep up w. tech | 7 | 7 | 2.124 |
| Large | Keep up w. tech | 6.65 | 7 | 2.206 |
| Small | Prominent. tech | 3.53 | 4 | 2.577 |
| Medium | Prominent. tech | 5.58 | 6 | 2.610 |
| Large | Prominent. tech | 5.76 | 5 | 2.029 |

## 5.2. Value Relations

As shown in Table 10, there are strong correlations (≥0.5, (Cohen 1988)) within the service, efficiency and engagement value positions, and also between the technocratic values. Citizen centricity strongly correlates with engagement values as well. Although the correlation within the professionalism paradigm is lower, legality and accountability correlate more with each other than with any other value. When running the same test on the municipalities and national agencies, respectively, correlation strengths similar to those in Table 10 were found.

**Table 10.** Correlations between values.

| Values | Legality | Accountability | Cost reduction | Productivity | Service and Quality | Citizen Centricity | Democratic Engagement | Participative Engagement | Keep up with Technology |
|---|---|---|---|---|---|---|---|---|---|
| Legality | - | - | | | | | | | |
| Accountability | 0.457 ** | | | | | | | | |
| Cost reduction | 0.113 * | 0.278 ** | - | | | | | | |
| Productivity | 0.212 ** | 0.277 ** | 0.509 ** | | | | | | |
| Service and quality | 0.170 ** | 0.249 ** | 0.190 ** | 0.346 ** | - | | | | |
| Citizen centricity | 0.193 ** | 0.332 ** | 0.277 ** | 0.378 ** | 0.565 ** | - | | | |
| Democratic engagement | 0.286 ** | 0.370 ** | 0.110 | 0.228 ** | 0.423 ** | 0.676 ** | - | | |
| Participative engagement | 0.196 ** | 0.408 ** | 0.255 ** | 0.301 ** | 0.445 ** | 0.683 ** | 0.797 ** | - | |
| Keep up with technology | 0.264 ** | 0.297 ** | 0.212 ** | 0.326 ** | 0.250 ** | 0.313 ** | 0.187 ** | 0.203 ** | - |
| Technological forefront | 0.273 ** | 0.324 ** | 0.148* | 0.204 ** | 0.192 ** | 0.254 ** | 0.222 ** | 0.191* | 0.604 ** |

\* Significant at $p > 0.05$; ** Significant at $p > 0.01$.

Tables 11–14 show the results of the component analysis. Table 11 assesses the sampling adequacy to 0.773, which was satisfying (the closer to 1, the better). Bartlett's test of sphericity had a low *p*-value (<0.001), which indicates that the data is suitable for a dimension reduction. The included variables are shown in Table 12, and they constitute a good fit since all extraction values were >0.5.

The three components explained 73.5 % of the variance in the data, as shown in Table 13. Finally, Table 14 describes the variables that are included in the three components (values <0.4 were removed

for greater readability). As can be seen, the analysis suggests that the data can be reduced to three dimensions: service and engagement values in Column 1, efficiency values in Column 2 and professionalism values in Column 3. Participative engagement, democratic engagement, and citizen centricity load >0.8 on the first component, while the service and quality item has a slightly lower load (0.663). Cost reduction and productivity represent the efficiency ideal under Component 2 with relatively high loads (>0.8), while legality and accountability load heavily on Component 3 with 0.855 and 0.755, respectively.

**Table 11.** KMO and Bartlett's test.

| Kaiser-Meyer-Olkin Measure of Sampling Adequacy. | | 0.773 |
|---|---|---|
| **Bartlett's Test of Sphericity** | Approx. Chi-Square | 756.366 |
| | Df | 28 |
| | Sig. | 0.000 |

**Table 12.** Communalities.

| | Initial | Extraction |
|---|---|---|
| Cost reduction | 1.000 | 0.763 |
| Service and quality | 1.000 | 0.512 |
| Productivity | 1.000 | 0.725 |
| Citizen centricity | 1.000 | 0.780 |
| Democratic engagement | 1.000 | 0.825 |
| Participative engagement | 1.000 | 0.792 |
| Legality | 1.000 | 0.791 |
| Accountability | 1.000 | 0.688 |

Extraction method—principal component analysis.

**Table 13.** Total variance explained.

| Component | Initial Eigenvalues | | | Extraction Sums of Squared Loadings | | | Rotation Sums of Squared Loadings | | |
|---|---|---|---|---|---|---|---|---|---|
| | Total | % of Variance | Cumulative % | Total | % of Variance | Cumulative % | Total | % of Variance | Cumulative % |
| 1 | 3.551 | 44.389 | 44.389 | 3.551 | 44.389 | 44.389 | 2.786 | 34.828 | 34.828 |
| 2 | 1.239 | 15.484 | 59.873 | 1.239 | 15.484 | 59.873 | 1.592 | 19.900 | 54.728 |
| 3 | 1.087 | 13.588 | 73.461 | 1.087 | 13.588 | 73.461 | 1.499 | 18.733 | 73.461 |
| 4 | 0.684 | 8.550 | 82.011 | | | | | | |
| 5 | 0.523 | 6.543 | 88.554 | | | | | | |
| 6 | 0.434 | 5.428 | 93.982 | | | | | | |
| 7 | 0.302 | 3.775 | 97.757 | | | | | | |
| 8 | 0.179 | 2.243 | 100.000 | | | | | | |

Extraction method—principal component analysis.

**Table 14.** Rotated component matrix [a].

| | Component | | |
|---|---|---|---|
| | 1 | 2 | 3 |
| Cost reduction | | 0.865 | |
| Service and quality | 0.663 | | |
| Productivity | | 0.806 | |
| Citizen centricity | 0.845 | | |
| Democratic engagement | 0.867 | | |
| Participative engagement | 0.861 | | |
| Legality | | | 0.885 |
| Accountability | | | 0.755 |

Extraction method—principal component analysis; rotation method—varimax with Kaiser normalization; [a]. rotation converged in five iterations.

*5.3. Examples of Value Operationalization*

In this section, initiatives found in the Dela Digitalt (2018) database are described, based on an interpretation of which value paradigm they can be positioned under.

Professionalism concerns adapting government systems and services to the European Union's General Data Protection Regulation (GDPR), effective 25 May 2018. As such, issues about treating user's personal data and following the requirements of the GDPR are widely discussed. Other topics in this area include public procurement, the use of electronic invoices (which is mandatory in the Swedish public sector), and e-archive.

Efficiency is represented in the database through evaluation and assessment of digital initiatives. The Swedish national financial management authority is actively promoting a framework for benefit realization. They encourage and educate government entities on how to assess and evaluate their digital solutions based on quantitative indicators. Their work is based on a task from central government to assess the cost of IT in the public sector. In addition, government entities are both asking for and providing examples of evaluations of their digitalization efforts. Another example of activities striving towards efficiency are the increased use of automation in different areas.

For service, electronic or digital services are common topics in the database, including descriptions of specific services. Many posts contain citizen-centric terms, shifting between referring to "digital self-service from a customer perspective", "user-driven", and "life-events". Other posts present inventories of services, and discuss how to integrate digital services with other channels, such as physical customer centers. One example from this material is a standardization of digital services for certain permits that are required for businesses. Municipalities can use these standards when procuring IT solutions, and the services can be integrated with a national business register.

In terms of engagement, few examples of direct participative engagement were found. One municipality is working with a digital forum to gather feedback from citizens in the development of a new web site. Other participatory initiatives include apps similar to "Fix-my-street" where citizens can report errors through geotagged photos. When searching "democracy" or "participation" in the database, the results revealed municipalities that focus on increasing digital inclusion by bridging digital divides through educational activities at schools, libraries, and in elderly care. An example is the introduction of tablets and virtual reality technology among elderly people. This activity was performed during the summer break and included temporary employment of adolescents who were responsible for teaching the elderly about these technologies.

Some technocracy initiatives are oriented towards the use of specific technologies. The Swedish Innovation Agency is looking to fund innovate uses of new technology such as Internet of Things in the public sector. One government actor is looking for proof of concept for robotic process automation in the public sector. Another post contains a query about the implementation of artificial intelligence in local and regional government.

## 6. Conclusions

The purpose of this paper was to investigate which values are prioritized in the Swedish digital government. By using categories of values from prior research in addition to technocratic values, a survey was constructed and administered to Swedish national agencies and municipalities. As such, the results of this research contribute to a growing body of research on value traditions associated with the implementation of technology in the public sector.

Three hypotheses were formulated and discussed below, in conjunction with the results from the interpretation of the qualitative data.

**H1:** *Value priorities differ based on organization type (national agency/municipality).*

(H0: There are no differences based on organization type.)

The results reveal that the municipalities grade all values except for professionalism and technocratic values significantly higher than the national agencies. Thus, the null hypothesis is rejected.

The municipalities also adapted more citizen-centric approaches, possibly because they are the government bodies that connect the most with citizens; for example, these agencies provide child care, education, social services and elderly care. Engagement values were prioritized notably low by the national agencies. Quite remarkably, these agencies prioritize keeping up with and being prominent in technology use ahead of including citizens in democratic and participatory processes. Accountability was given relatively low priority in the material, which deserves attention in further research. It should be noted, however, that accountability is a rather complex term that might be difficult to translate and interpret through a survey. Some of the variances found can be explained by the variety of organizations and areas of responsibility. National agencies have a higher standard deviation than municipalities, which can be explained by more specialized areas of responsibility.

**H2:** *Value priorities differ based on organization size.*

(H0: There are no differences based on organization size.)

No differences based on the number of citizens in the municipalities were found. Among the national agencies, the largest agencies prioritized efficiency and technocracy values more than the other agencies. Hence, the null hypothesis is accepted for the municipalities and rejected for the national agencies. A plausible explanation for the differences based on organization size is that small organizations rely less on technology than larger and have limited potential for increased efficiency. In some cases the smallest agencies rely on the larger agencies in shared hosting solutions.

**H3:** *Values can be divided into four distinct positions (professionalism, efficiency, service, and engagement).*

(H0: Values cannot be divided into four distinct positions.)

The results of this study suggest that professionalism and efficiency are distinct positions, while service and engagement are closely related through citizen centricity. The null hypothesis is accepted.

The combined findings from the quantitative and qualitative data suggest that the agencies in the Swedish digital government, which can be more-or-less citizen-centric, can be described as (e)-service producers within regulated environments. Service and quality, productivity and legality were the main drivers:

- Service and quality were manifested by a variety of digital services in different areas. This is not surprising considering these services constitute the means to facilitate communication and transactions between government entities and civil society.
- Efforts to increase productivity, for example by automation, were subject to queries about how to evaluate and assess the outcomes.
- Legality was represented by adapting work with digital government to new laws, such as the GDPR, but also by electronic archives and routines for the procurement of IT.

Moreover, the results suggested that citizen centricity is an ambiguous term which can be manifested in various ways. One way is through the creation of services based on perceptions of the life events of the individual rather than organizational structures. In this service logic, the citizen is often referred to as a customer, or user, whose demands can be satisfied through a supply of digital services. While few examples of direct participatory activities were found in the material, another example of citizen centricity was activities of digital inclusion, with the aim to increase the digital literacy of vulnerable groups. Hence, a suggestion for future studies is to further refine the concept of citizen centricity in relation to digital government and public values.

*Limitations*

This study is not without limitations. The results are limited to the Swedish digital government context. A limited selection of values was used in this study and further research should study additional values and their relations. E-Government and digitalization are topics sensitive to trends, which might influence the results. For example, several government entities were working on adapting their systems to the GDPR during the survey, which might affect the ranking of legality.

**Funding:** This research received no external funding.

**Acknowledgments:** The author wishes to thank the respondents from the Swedish national agencies, and municipalities, who contributed their time to the study.

**Conflicts of Interest:** The author declares no conflicts of interest.

## Appendix A. Survey

Type of organization
1A National agency
1B Municipality

If 1A: Number of employees at the agency (estimated)
If 1B: Population in the municipality (estimated)

We digitalize to …
(In random order), 0–10, where 0 = not at all prioritized and 10 = highest priority.

… act according to current laws.
… clarify responsibility and legitimacy in decision making.
… reduce our costs.
… increase our productivity.
… increase service and quality for citizens and companies.
… put the citizen in the center.
… include the citizens in democratic processes.
… include the citizens in decision making processes.
… keep up with technology.
… be at the technological forefront.

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
