# Peer review of "Value Positions and Relationships in the Swedish Digital Government"

_admsci, doi:10.3390/admsci9010024_

Round 1
Reviewer 1 Report
The manuscript examines the value positions of employees in Swedish municipalities and public agencies. The literature review and the short history of values in the Swedish e-government context seems appropriate. The original contribution of the manuscript is to look at differences between municipalities and agencies. The manuscript is well written and well structured, but has some flaws described below.
1 Introduction
In Section 1 the authors claim that the empirical material consists of perceptions from public administration employees. But it is also said that the survey was administered to Swedish municipalities and national agencies. The sample size indicates that the survey was sent to the official e-mail address of each municipality and agency. It is unclear who answered the survey. It may be differences between management level and employees. The authors should explain how they made sure the survey was distributed to employees who work with value production daily. An alternative is to say the survey was answered by official representatives of the municipalities and agencies.
3 Values in the Swedish e-Government context: A short history
Page 5 line 180: Acronym NAO should be defined when first used (even if it is part of the reference). In 2004, the National Audit Office (NAO) concluded…
4 Materials and methods
The authors should explain why (only) two values definitions were chosen from each value position (from Rose et al. 2015:1).
In Table 2:
The value definitions should be corrected as follows:
Service and quality -> Service level and quality
Participatory engagement -> Participative engagement
Page 6 line 244: Rose et al. (2015), ambiguous. Either (2015:1) or (2015:2)
5 Results
References
Figure 1 shows a plot of the values from Table 13, but the explanation of Figure 1 could be better. Why is the figure necessary, and why does it present the results in a better way than the table?
Page 12 line 354. The authors earlier said municipalities were grouped based on number of citizens, not employees. (Page 5 line 200).
Author Response
Dear reviewer,
Thank you for the constructive feedback on my paper.
Please see the attached PDF for response to the review.

Reviewer 2 Report
The paper was well-structured. Statistic info and graphical illustration is clearly presented.
Few concerns before move to publication:
Try to provide a more detailed description or connection about the four main values in the part of lit review
Pay attention to the qualitative part in the research. Try to elaborate it with more details and supporting data
For the results of quantitative part, the reporting is quite brief in some sections. It is expected to highlight some specific significant figures in the paragraph (eg. the correlation between values)
A more in-depth discussion should be included in the paper
A thorough cross-check of all references is suggested for accuracy and formatting, especially the part of author and the year of publication.
Author Response
Dear reviewer,
Thank you for the constructive feedback on my paper.
Please see the attached PDF for my response on the review.
